# Trauma-Responsive Vocational Rehabilitation Services

**DOI:** 10.3390/bs13060511

**Published:** 2023-06-19

**Authors:** Sara Chopp, Dimitri Topitzes, Joshua Mersky

**Affiliations:** Helen Bader School of Social Welfare, University of Wisconsin-Milwaukee, 2400 E Hartford Ave, Milwaukee, WI 53211, USA; topitzes@uwm.edu (D.T.); mersky@uwm.edu (J.M.)

**Keywords:** trauma-informed care, trauma-responsive services, culturally-responsive practices, racially equitable service delivery, strengths-based practices

## Abstract

Research on the effectiveness of Vocational Rehabilitation (VR) Programs has revealed that VR services are less effective for trauma-affected and Black consumers. For instance, consumers with trauma exposure disengage from services earlier than their non-traumatized counterparts, and Black consumers benefit less from each phase of VR services compared to others. One midwestern state’s VR program sought to address these disparities by offering trauma-informed and trauma-responsive services that emphasize cultural responsiveness, racial equity, and strengths-based practices. To begin this work, the state’s VR program collaborated with an applied research unit in a public university to establish two work groups: a communications group and a training group. The purpose of the communications group was to build a robust referral network within the VR Division and with other community-based agencies and providers, particularly for low-income, Black consumers. The purpose of the training group was to develop and deliver a training program to support VR professionals in providing trauma-informed and trauma-responsive services. Results from an evaluation of the training indicated that each training module generated for staff both reminders and new insights into ways to effectively work with consumers. Staff expressed that they wanted opportunities to further explore and apply the training content and needed additional, ongoing support to implement what they were learning. In response to staff needs, the state’s VR program is continuing to invest in this community–university partnership by establishing communities of practice for staff and evaluating the effectiveness of the training program.

## 1. Introduction

State vocational rehabilitation (VR) services programs provide individualized VR services to people with disabilities, “so that they may prepare for and engage in competitive integrated employment or supported employment and achieve economic self-sufficiency” [1]. These programs offer a menu of services that includes assessment, information and referral services, job training, education, job search assistance, and on-the-job support [1]. Program evaluations indicate that VR services can help people with disabilities attain higher levels of education, competitive employment, and higher pay [2]. Evaluation research also suggests that VR services can enhance consumers’ psychosocial functioning along with mental and physical health [3].

However, researchers have revealed race-based differences in all dimensions of the VR process: application, acceptance, service provision, closure status, and vocational outcomes. Studies have shown that Black Americans are less likely to apply for the VR services program than their White counterparts [4]. Black Americans who do apply are accepted into the VR service program at a disproportionately lower rate than White Americans [5]. Compared to other groups, Black Americans who are accepted into the VR services program receive fewer services, receive different patterns and frequencies of services, have less money spent on them, and spend less time in the VR program—resulting in poorer vocational outcomes such as employment and earnings [6,7,8,9,10].

To foster greater equity throughout the VR services program both in terms of access and outcomes, one midwestern state’s Division of Vocational Rehabilitation (DVR) has been working to improve service delivery to low-income, Black consumers. The primary aim of the project was for the DVR to identify weaknesses that inhibit its ability to cultivate engagement in and positive outcomes for low-income, Black consumers. Identified program limitations included a lack of trauma- and culturally responsive services and a lack of racial equity throughout the VR process. The DVR partnered with a state university to develop trauma- and culturally responsive programming with the help of communications and training groups. The former worked to improve access to and support from VR services for low-income, Black consumers. The latter developed and delivered training for staff in trauma and culturally responsive, racially equitable, and strengths-based services.

The following sections are organized as follows. First, we will summarize the results from a qualitative evaluation that was conducted with consumers and staff from a DVR agency in a midwestern city as part of a quality improvement project. Subsequently, we will use empirical evidence to discuss the themes and support the recommendations that came out of that evaluation. Second, we will describe how these data and recommendations informed the development of a training program for DVR staff, and we will outline each training module of the curriculum. Third, we will discuss the results of evaluations completed by participants after the delivery of each training module. Finally, we will conclude with a discussion on the implications of our findings and future directions.

## 2. Moving toward Trauma- and Culturally Responsive Services: A Multi-Year Initiative

### 2.1. Phase I: Uplifting the Voice of DVR Consumers

Recognizing the disparities in workforce development that have been observed nationally and locally, a midwestern state’s DVR launched a multi-year initiative to improve the quality of services delivered to low-income, Black consumers. In May 2019, the DVR and a consulting agency conducted focus groups with 30 low-income, Black consumers of DVR services to learn about their experiences with DVR services and their barriers to employment. Consumers suggested that the DVR can improve its marketing and outreach to better provide resources and opportunities for those living in their area. They also suggested that the DVR can strengthen collaboration across programs, partners, and employers to improve services and outcomes for low-income, Black consumers. Consumers also wanted the DVR to hire more staff. Consumers expressed concerns about inconsistent staffing, counselor turnover and case transfers, and counselor stress and caseloads [11].

Consumers described race- and poverty-related barriers they have experienced and a need for counselors to have empathy regarding these challenges. Consumers wanted counselors to actively listen, spend more time with them, and be more flexible in how they meet and communicate with them. Consumers also expressed a need for counselor follow-through and follow-up and emphasized the importance of receiving clear communication from responsive counselors and having a strong working alliance with counselors. Consumers preferred counselors who are knowledgeable about educational and vocational opportunities and provide consumers with direct connections to those opportunities. Consumers expressed a strong desire to work, improve their lives, and create a better future for their children [11].

By conducting the focus groups with low-income, Black consumers, the DVR was able to identify factors that limit its capacity to reach, engage, and support this service group. In conclusion, the main concerns unveiled were a lack of trauma-informed and culturally responsive services and a lack of racial equity for consumers. This makes sense when considering that individuals with disabilities are at greater risk for trauma exposure compared to the general public [12], and Black Americans with disabilities—especially those who live in poverty—have an even greater chance of being exposed to trauma, experiencing compounded effects of trauma [13] and experiencing multiple forms of trauma [14]. Individuals with extensive trauma histories may be more likely to have aversive experiences with VR practices and processes as service engagement can trigger trauma memories, eliciting fear and defensive behaviors [13]. Consumers with trauma histories may be more sensitive to the timeliness of VR service delivery, providers’ withholding of information about the VR process, discrimination in service delivery (e.g., equity in terms of time and money spent and effectiveness of types of services received), and cultural barriers [13,15,16,17] This is consistent with the themes that emerged during the focus groups as detailed above.

Black Americans may also experience institutional trauma that fosters cultural mistrust towards White Americans and institutions due to historical race-related mistreatment [13,18,19,20]. Cultural mistrust among Black VR consumers may be activated by experiences of institutional bias or discrimination [21]. Consumers experiencing cultural mistrust may shun or prematurely withdraw from the VR services program, be reluctant to engage in the VR services program, hold negative attitudes about seeking help from VR services programs that are staffed primarily by White Americans, and limit self-disclosures to White American counselors [18,19,20]. Therefore, cultural mistrust resulting from institutional trauma may be contributing to racial inequities throughout the VR services program.

Organizational factors can also influence consumers’ vocational outcomes, effects that may be magnified for consumers with trauma histories. VR staff are at high risk for burnout and turnover due to systemic factors such as administrative demands, heavy workloads, and time pressure [14]. It is also common for VR staff to experience high levels of job-related stress associated with difficult consumer behaviors, including verbal and physical assaults, and they may also be at risk of secondary traumatic stress due to frequent firsthand exposure to consumers’ trauma histories [22]. These conditions are known to contribute to staff turnover, low-quality services (e.g., rushing, dismissing, and patronizing consumers), and poor outcomes for consumers with trauma histories [16,22,23,24]. These factors emerged as opportunities for improvement during the focus groups as detailed above.

Conversely, research suggests that consumers—particularly those with trauma histories—benefit appreciably when they have meaningful relationships with VR professionals [25]. Prevailing theory and evidence suggest that meaningful consumer-professional relationships are more likely to emerge when VR staff are: (a) physically, psychologically, and emotionally present; (b) culturally sensitive and altruistic; and (c) interact with consumers in a positive and considerate manner [16,25,26]. Chang (2018) also found that creating a context of safety, trust, transparency, autonomy, and acceptance promoted positive vocational, psychosocial, financial, and legal outcomes for consumers. Rooted in these trauma-informed and strengths-oriented principles, VR counselors can foster positive vocational outcomes by establishing a rapport and strong working alliances with consumers while also maintaining high expectations and emphasizing the importance of follow-through [16]. In summary, embedding trauma-informed principles into programming has the potential to enhance interactions and VR agency culture in ways that promote positive outcomes for consumers and staff alike [27].

While trauma-informed and trauma-responsive services may benefit all VR consumers, they could particularly help Black consumers given this group’s disproportionate exposure to trauma and historical experience with VR services. Practices that are well-tailored for trauma-affected consumers compared to traditional services can help create an organizational culture that is more strongly relationship-based and more sensitive to consumer needs. In turn, program engagement and outcomes may improve [14,27,28]. To improve service delivery to low-income, Black consumers, the DVR is integrating trauma-informed care principles [29] and implementing trauma-responsive practices [30] with an emphasis on cultural responsiveness, racially equity, and strengths-based practice. In the following sections, we will describe the development of this program, including the activities of the communications and training groups.

### 2.2. Phase II: Developing a Trauma- and Culturally Responsive Program

In partnership with a local university, the DVR has taken steps to create a trauma-responsive, culturally responsive, racially equitable, and strengths-based organization for staff and consumers by forming two specific work teams: a communications group and a training group. The communications group, which consisted of both university and DVR staff, was designed to establish a robust referral-in and referral-out network for DVR consumers. A referral-in network increases awareness of and access to the VR services program, particularly among low-income, Black consumers. The referral-out network helps the DVR connect consumers with the external resources needed to promote favorable VR service engagement and outcomes.

The training group also comprised university and DVR staff including field staff, supervisors, and senior leadership, some of whom were individuals with disabilities and people of color. The training group developed a training program for staff designed to enhance the trauma-responsive, culturally responsive, racially equitable, and strengths-based nature of VR services. Becoming more trauma-responsive involves understanding different types of trauma, developing an awareness of trauma prevalence, gaining insight into trauma consequences, and knowing how to best respond to trauma-affected consumers. Becoming more culturally responsive and racially equitable results in effective approaches to mitigating inequities and inequalities. Becoming strengths-based involves identifying and cultivating consumers’ internal and external resources that promote well-being, including educational and vocational success. A total of eight modules were developed and delivered by university personnel with support from DVR staff over the course of eight months. Training modules were delivered once per month during all-staff meetings lasting approximately two hours. The following sections specify the content that was delivered during each session, why it was delivered, and how it was delivered. Table 1 provides an overview of module content, activities, and aims.

#### 2.2.1. Module One: Introduction

The first training session was designed to demonstrate the difference between the traditional model of service delivery and one that is trauma-responsive, culturally responsive, racially equitable, and strengths-based. In collaboration with participants, the university program staff (first and second paper authors) established group norms for engaging in the training and group agreements for how to engage with each other throughout the training. The trainers introduced Multnomah County’s (2022) “Challenge Circles”. The “Challenge Circles” include three zones: the comfort zone, the challenge zone, and the danger zone. When we’re in our comfort zone, we are calm and complacent; when we are in our challenge zone, we are excited and challenged; and when we’re in our danger zone, we are stressed and overwhelmed. The trainers guided the participants in reflecting on what it felt like to be in each zone. The challenge zone is where we learn and grow, so the trainers encouraged participants to stay in the challenge zone and to do whatever they needed to do to return to the challenge zone if they found themselves in the danger zone. The trainers also set standards and expectations for communication: using I-messages, active listening, compassionate assertiveness, and nonviolent communication. At the beginning of each subsequent training session, the trainers reminded participants about the group norms, group agreements, and standards for communication.

After the first training session, participants were provided an opportunity to complete a survey that contained Diemer et al.’s (2017) Critical Consciousness Scale, Neff’s (2011) Self-Compassion Scale, and a modified version of the Schwartz Values Survey that isolated benevolence and universalism values. Participants were also asked to complete a standard self-affirmation intervention with the value they selected as most important. Self-affirmation interventions have been found to boost individuals’ sense of responsibility for behaving in ways that express a concern for the welfare of others and humanity and their willingness to do so [53,54] and they have been found to enhance individuals’ self-concept [55]. This approach was used to encourage participants to reflect on their most important core personal value and a time during which that value was important [33]. Doing so, in turn, may reduce self-serving biases that contribute to harmful attitudes and behaviors such as racial microaggressions and erroneous individual attributions such as victim blaming [56]. Reducing self-serving or defensive bias can also lead to altered internalized norms, attitudes, and beliefs, bringing them in closer alignment with values such as benevolence and universalism that are motivated by a concern for the welfare of others [57].

#### 2.2.2. Module Two: Trauma-Responsive Practice

The trainers presented on the following topics during the second session: definitions of trauma, prevalence of trauma exposure, consequences of trauma exposure, and service frameworks tailored for trauma-exposed individuals such as trauma-informed care and trauma-responsive practice. The discussion of trauma definitions covered acute and chronic trauma, individual and collective trauma, and historical and systemic trauma. Content on trauma prevalence reflected epidemiological data published in local, state, and national studies. Finally, the trainers delivered information on trauma service frameworks that can be used to guide practice [34,35].

The trainers delivered the above-mentioned information didactically but also facilitated two interactive discussions. First, they asked participants to reflect and report on their understanding of trauma before and after the discussion of trauma definitions. Second and prior to introducing trauma service frameworks, the presenters introduced a case example developed by DVR staff and prompted small group discussions around trauma-responsive services for the case. Each small group was led by a designated facilitator from the trauma training workgroup.

#### 2.2.3. Module Three: Critical Reflection

The third training session was designed to facilitate staff engagement in critical reflection and to support their development of critical motivation. First, the trainers introduced Bronfenbrenner’s (1979) ecological systems model to help participants gain an understanding of the systems in which we live and how transactions with those systems enable and inhibit our well-being. After the trainers presented this material, they had participants reflect on their own ecological niche. Then, the trainers provided a high-level overview of the historical antecedents to the collective ecological systems using Native American history and Black American histories. The trainers connected the historical, collective experience of Native Americans and Black Americans to historic, cultural, systemic, and interpersonal trauma. They focused on inequities experienced by Black Americans in the city of focus and how those inequities produce inequalities of well-being. Then, the trainers presented inequities that exist throughout the VR services programs and how the inequities produce inequalities in outcomes for Black consumers. The trainers created small groups in which participants developed an eco-web for a consumer that was described in the case study [40].

In the second half of this training session, the trainers taught participants about different types of attributions used to explain the cause of events. They demonstrated how one can either blame consumers for the cause of their adversities and lack of engagement in and negative outcomes from the VR services program or recognize forces that create and maintain inequities for particular consumer groups [37,38] The trainers connected attributions to more general beliefs about inequities and inequalities, namely beliefs in a natural social hierarchy versus egalitarianism [38,39]. The trainers encouraged participants to endorse egalitarian beliefs and align their attributions with them. The trainers asked participants to self-reflect on their responses to Diemer et al.’s (2017) Critical Consciousness Scale, which includes measures of attributions for and beliefs about inequities and inequalities. The trainers also asked participants why their attributions and beliefs did or did not change after learning more about context.

#### 2.2.4. Module Four: Trauma-Responsive and Strengths-Based Practices

The fourth training session was designed to teach participants about trauma-responsive and strengths-based practices. In the first half of this training session, the trainers presented the elements of well-being: positive emotions, engagement, positive relationships, meaning, and achievement [58]. They also presented the core and additional features that individuals typically need to flourish. The core features include positive emotion, engagement, and meaning, with the additional features being self-esteem, optimism, resilience, vitality, self-determination, and positive relationships [58]. Additional content on self-compassion was covered in the eighth training session (see below).

The trainers explained how character strengths are the underpinnings of well-being [58] and defined each of the six virtues under which 24 character strengths can be organized. They explained the relationships between each virtue and described how character strengths can be ignited in various contexts [40,59]. The trainers requested participants complete Peterson and Seligman’s (2004) VIA Character Strengths Survey prior to the fourth training session to make the content more meaningful, and so that the trainers could ask the participants to complete a strengths affirmation after the trainers presented the character strengths content. For the strengths affirmation, the trainers asked participants to choose one of their top five strengths of character, to think about how that strength describes the “real them”, why that strength is important to them, and in what way that strength serves them. Then, the trainers placed participants into small groups in which they discussed the ways they could use their character strengths in their work with consumers and how they could help consumers ignite their own character strengths at work and at school to improve their engagement in and outcomes from the VR services program.

The second half of the training session built upon session two, exploring in depth the assumptions and principles of trauma-informed care (TIC) along with the practices known as trauma-responsive. More importantly, the trainers applied constructs presented didactically in this session to the case example introduced in session two. Specifically, after the trainers detailed a particular concept, i.e., an assumption and principle of TIC, they provided an example of how to operationalize the concept when working with the adult identified in the case. Subsequently, the trainers organized participants into smaller groups, each with a designated and well-prepared facilitator, and prompted each group to apply TIC principles to the case.

#### 2.2.5. Module Five: Mindfulness and Optimistic Explanatory Style

The fifth training session was designed to teach participants about mindfulness and an optimistic explanatory style and to support their development of critical motivation. In the first half of this training session, the trainers presented the definitions and benefits of the three pillars of mindfulness: intention, attention, and attitude [41]. They led participants through a guided mindfulness meditation [41] and then placed participants in small groups in which participants were asked to discuss how they could use intention, attention, and an attitude of kindness and curiosity in their work with consumers. The trainers encouraged participants to use the value they selected in module 4 as the most important for their intention.

In the second half of this training session, the trainers presented the definitions and consequences of optimistic and pessimistic explanatory styles [42]. Prior to this training session, the trainers asked participants to complete Seligman’s (2006) Optimism Test to make the content more meaningful and to facilitate participant engagement in the discussion. The trainers placed participants in small groups in which they asked them to use Seligman’s (2006) ABCDE method to explore how they think about a consumer’s case in terms of explanatory style and to shift their thinking to be more optimistic.

#### 2.2.6. Module Six: Growth Mindset and Grit

The sixth training session was designed to teach participants how to cultivate a growth mindset and grit to support their development of critical motivation. In the first half of the training session, the trainers presented the definitions and consequences of fixed and growth mindsets [44]. After the trainers presented the content on mindsets, they asked participants to self-reflect on a time when they received feedback from someone else about a mistake they made, whether they responded with a fixed or growth mindset, and how they could enhance their growth mindset in the future. The trainers also asked participants to self-reflect on a time when they gave feedback to someone else about a mistake that person made, whether participants responded with a fixed or growth mindset, and how they could respond with more of a growth mindset in the future.

In the second half of this training session, the trainers presented the definition and consequences of grit [45]. Grit is comprised of two components: passion and perseverance. Passion involves identifying that which is most important to you—your ultimate concern or compass—and staying committed to that goal over time. Perseverance involves pushing hard and continuing to push to turn your ultimate concern about, despite challenges. The trainers described how one can use one’s most important personal value as a compass and construct a goal hierarchy in a way that supports the expression of that most important value [45]. The trainers encouraged participants to use the value they identified as the most important when they completed the Schwartz Values Survey as their top-level goal. Then, the trainers asked participants to construct their goal hierarchy to be in alignment with that value [45]. The trainers placed participants in small groups in which the trainers requested that participants discuss how they could connect their work with consumers to their most important value.

#### 2.2.7. Module Seven: Trauma-Responsive Practices, Continued

Building on previous sessions, particularly two and four, session seven focused on trauma-responsive practices. The trainers reminded participants of trauma-responsive practices— such as screening and assessment, information giving, and referral—and applied each to the case example introduced in session two. Subsequently, the trainers directed participants to their small groups and them to consider how they would use trauma-responsive practices while providing VR services to the case. Finally, all participants returned to the larger training setting and discussed, as a group, barriers to implementing trauma-responsive practices within the VR context and solutions to overcoming these barriers.

#### 2.2.8. Module Eight: Critical Consciousness and Self- and Collective Care

The eighth training session was designed to facilitate participants’ engagement in critical consciousness and to support their self- and collective care. In the first half of this training session, the trainers presented the definition and consequences of critical consciousness [36,39,46,47,48,49,50,51] The trainers connected each component of critical consciousness to the positive psychological principles and practices they delivered throughout the training program. As a large group, the trainers encouraged participants to brainstorm critical actions they could take at multiple levels directed at multiple targets to protect and promote the welfare of themselves, others, and humanity and to stop harm and exploitation of themselves, others, and humanity.

In the second half of this training session, the trainers presented the definition of self-compassion and its consequences [52]. As a large group, the trainers encouraged participants to share their self-reflection on how external influences affect how they think and feel about themselves, how external influences affect the way they think and feel about others, and what practices and skills they could use to support strengths-based thoughts about themselves and others [31]. Then, the trainers asked participants to share thoughts they have about themselves that help them get through tough times, what thoughts get in the way of placing themselves as a top priority, and what skills and practices they can use to change inhibiting thoughts [31]. Finally, the trainers presented the definition and consequences of collective care [31]. The trainers also asked participants to share examples of collective care they have witnessed or learned about.

## 3. Results

### Module Evaluations

Following the delivery of each module, participants received a link to a Qualtrics survey to provide feedback on the training session that asked the following questions: (1) What is one thing you see differently after this workshop? (2) What is one thing you learned that you’ll remember from this workshop? (3) What is one thing that made you happy from this workshop? (4) What is one thing you would change or reshape from this workshop?

Generally, the 18 participants who provided feedback on the introduction module (1) expressed that they learned new communication skills and enjoyed interacting with their coworkers. The 12 participants who provided feedback on the mindfulness and optimistic explanatory style module (5) mentioned that they learned how to change the way they think to be more optimistic and new ways of thinking about and interacting with consumers. They suggested they were reminded of the importance of practicing mindfulness. They enjoyed interacting with their coworkers and felt a sense of common humanity with them. They wanted more time for small group discussions and to dive deeper into the content. The six participants who provided feedback on the growth mindset and grit module (6) said they learned about their own mindsets and how they can affect the mindsets of others. They reported they would remember what they learned about their mindsets and how to align goals with their most important value. They enjoyed sharing their thoughts, reflecting on their values and goals, and engaging in group discussions with their coworkers. They wanted more time to discuss the content with their coworkers.

The 16 participants who provided feedback on the trauma-responsive practice module (2) expressed that they learned about multiple forms of trauma, various causes of trauma, and how trauma affects people’s behaviors. They wanted to know more about how trauma-responsive practices can be used with existing DVR policies and processes. They enjoyed hearing different perspectives from their coworkers and wanted more opportunities for small group discussions. The five participants who provided feedback on the trauma-responsive and strengths-based practices module (4) indicated that they learned about themselves and their coworkers. They expressed a desire to use strengths-based and trauma-informed principles with consumers. They enjoyed discussing the content with their coworkers. They learned about the positive ways their coworkers think about them, and they felt that the room was filled with positivity. The 17 participants who provided feedback on the second trauma-responsive practice module (7) said they were reminded of how to use trauma-responsive practices with consumers and that it affirmed what they were already doing with their consumers. They enjoyed sharing their insights and interacting with their coworkers. They wanted more time to discuss the content with their coworkers. They expressed frustration with the suggestion that they should be referring consumers out to other services to support their engagement in and improve their outcomes from the VR services program. Staff already feel overwhelmed with heavy caseloads and lack of support and don’t see how they can do more than what they are already doing.

The few participants who provided feedback on the critical reflection module (3) expressed a need for more internal and external resources to effectively serve their consumers. They were frustrated with a perceived lack of support available to them and great demands placed on them while serving large caseloads. They were happy about their interactions with coworkers, felt stronger solidarity with them, and wanted to dive deeper into the content. The 13 participants who provided feedback on the critical consciousness and self- and collective care module (8) suggested that they had a better understanding of actions they could take (small and large) at multiple levels (including within themselves) to create change within and outside of the VR system. They were reminded of the importance of self-care and learned how to practice self-compassion. They enjoyed learning more about their coworkers and experiencing common humanity with them. They appreciated their coworkers’ willingness to share their life experiences, their resources, and actions that they’re currently taking to create change. They wanted to dive deeper into the content and more support (e.g., walking breaks) to digest the content. They requested more activities and discussions to reinforce the content in between training sessions.

## 4. Future Directions

### 4.1. Training and Consultation

With support from the university, the DVR will establish Communities of Practice (COPs) to deliver ongoing technical assistance and consultation to help staff implement what they learned in the training series described above. COPs activities will anchor on the topics of trauma-informed and -responsive practices and include case review, role play, question and answer, and thematic discussion. While revisiting topics discussed during the training series, the COPs will also introduce novel and advanced content to help engage staff in activities and motivate them to apply the material to their professional service.

The COPs will be led by university personnel along with DVR staff who select into the group based on stated interest in the educational and vocational success of low-income, Black consumers. This group will meet regularly to prepare for COPs sessions by developing an agenda and activities, reviewing training and educational content, and developing new and advanced material. Along with trauma-related foci, the COPs leaders will also integrate material with the explicit intent of improving racial equity throughout and racial equality in outcomes from the VR services program. The shape of ongoing training for new staff has yet to be determined, but staff interests and needs along with qualitative evaluation of training modules will help determine new staff training directions.

Regardless of the specifics, the DVR and the university intend to continue to train new DVR staff in trauma-responsive, culturally responsive, racially equitable, and strengths-based services. In addition, the university will help the DVR establish supports for staff who are experiencing vicarious trauma. The university will also develop and deliver training and resources to the DVR’s Business Service Consultants, who help link DVR consumers with outside employers. By working with the DVR’s Business Services Consultants along with outside employers, the DVR–university partners hope to enrich the services that the consultants provide DVR consumers. There is also a budding trauma-informed employment movement that the DVR–university partners plan to invoke when interfacing with DVR partner employers such as Goodwill Industries. The ultimate goal, of course, is to strengthen service models for DVR consumers and promote favorable employment outcomes, particularly among trauma-affected and Black American consumers. (e.g., pay per hour and hours worked).

### 4.2. Evaluation

The midwestern state’s DVR is at the beginning of its organizational change process. There are several areas that the DVR can focus on to broaden and deepen trauma responsiveness, cultural responsiveness, racial equity, and strengths-based practice throughout the organization. To do this, the DVR will continue to work with the university to establish accurate data systems for measuring compliance with the consumer service improvement plan including measures to assess progress towards each outcome domain (i.e., trauma-informed and -responsive, culturally responsive, racially equitable, and strengths-based) and for assessing formative and summative outcomes for staff and consumers including robust staff and consumer satisfaction measures.

### 4.3. Dissemination and Integration

If successful, the work being done through this university–community partnership could be used as a model for other state DVR agencies across the nation. The training program can potentially help VR professionals improve consumer engagement in and outcomes from the VR services program by developing their knowledge of trauma types, increasing their understanding of how trauma impacts the lives of consumers, and learning how and receive support to implement trauma-informed and trauma-responsive approaches in service delivery. To improve consumer engagement and outcomes and prevent and intervene in staff burnout, the VR services program as an organization can ensure the physical and emotional safety of staff and consumers; maximize trustworthiness between staff across all levels and between staff and consumers; maximize choice and control for staff and consumers; maximize collaboration and sharing power between staff across all levels and between staff and consumers; prioritize empowerment and skill-building for staff and consumers; enable peer support and mutual self-help for consumers; and consider cultural, historical, and gender issues [29,60].

A major component of the trauma-informed care approach is taking into consideration cultural, historical, and gender issues [29]. This involves identifying oppressive systemic forces (e.g., societal discourses and laws), systemic obstacles (e.g., Medicaid, Medicare, and social security disability programs), and historical contexts (e.g., colonialism and personal, systemic, and intergenerational trauma) that influence consumer engagement in and outcomes from the VR services program [61]. The training program could support DVR agencies to tailor VR services for different consumer groups to enhance their engagement in and outcomes from the VR services program. It could support VR professionals to overcome systemic biases related to race and other consumer characteristics that affect consumer engagement in and outcomes from the VR services program. For example, VR professionals could identify systemic inequities that cause inequalities in VR outcomes for Black consumers and connect Black consumers with additional services and resources to produce greater equality in VR consumer outcomes [6]. It could also support VR professionals to overcome cultural biases related to race and other consumer characteristics that contribute directly to inequities throughout the VR services program. For example, VR professionals could acquire specific skills for working with Black consumers and enhance the way they interact with Black consumers [62]. They could also identify inequities at all major junctures in the VR process and target those points for intervention to create change. We recognize that other systemic actions, e.g., hiring specialists, would likely be necessary to make a sustainable and widespread change.

In addition, the VR services program has data systems that track the number of applications received; the number of consumers accepted; the type, number, frequency, and pattern of services provided; the amount of money spent providing services; the amount of time spent in the VR services program; and closure status and quality of closure, and vocational outcomes in terms of types of employment (i.e., competitive versus non-competitive), pay per hour, and hours worked and disaggregating that data by race. This data system could be used to identify and track changes in inequities throughout and inequalities in outcomes from the VR services program. When inequities and inequalities are reduced, efforts can be positively reinforced. When inequities and inequalities are not reduced or are increased, efforts can be modified and additional reinforcements can be used. This data system includes consumer satisfaction surveys that could allow VR professionals to identify consumers at risk of unsuccessful closure and immediately intervene with well-tailored services and resources. Survey results could also help identify cultural biases and cultural insensitivities among VR counselors and inform corrective inputs such as increased supervision, education, and in-service training [6].

Results from the data system could be used to validate the effectiveness of the work one midwestern state’s DVR agency is doing to improve engagement in and outcomes from the VR services program for low-income, Black consumers. Then, the consumer improvement plan including the training program and specific data system usage could be adopted by and tailored for other state DVR agencies who are interested in reducing racial inequities and inequalities.

## Figures and Tables

**Table 1 behavsci-13-00511-t001:** Module Content, Activities, and Aims.

Module	Content	Activities	Aims
1	Introduction Group agreements Challenge Circles [31] I-messages Active listening Compassionate assertiveness Non-violent communication	Diemer et al.’s (2017) Critical Consciousness Scale Neff’s (2011) Self-Compassion Scale A modified version of the Schwartz Values Survey [32] A standard self-affirmation intervention [33]	Understand the difference between the traditional model of service delivery and trauma-responsive, culturally responsive, racially equitable, and strengths-based. Group norms Self-regulation Standards for communication Boost sense of responsibility for the welfare of others Boost willingness to contribute to the welfare of others Enhance positive self-concept Reduce self-serving or defensive biases
2	Trauma-Responsive Practice	Reflect on and report definitions of trauma [34] Small group discussion around trauma-responsive services for a case [35]	Understand the definitions of trauma, prevalence of trauma exposure, and consequences of trauma exposure Apply trauma-informed care and trauma-responsive practice to a specific case
3	Bronfenbrenner’s (1979) ecological systems model Historical antecedents to ecological system Systemic inequities Institutional inequities Causal attributions [36,37,38] Beliefs about inequities and inequalities [38,39]	Self-reflection on ecological niche Eco-web for consumer in the case study [40] Self-reflection on responses to Diemer et al.’s (2017) Critical Consciousness Scale	Identify forces that enable and inhibit well-being Connect historical and collective experiences to various forms of trauma Connect systemic inequities to inequalities in well-being Connect inequities in service delivery to inequalities in consumer outcomes Switch victim-blame attributions for system-blame attributions Switch beliefs in a natural social hierarchy to egalitarian beliefs Align attitudes and beliefs with a concern for the welfare of others
4	Seligman’s (2013) Well-being Theory Peterson and Seligman’s (2004) Character Strengths Trauma-Informed Care and Practices	Peterson and Seligman’s (2004) VIA Character Strengths Survey Strengths affirmation Small group discussion about using character strengths at work [41] Large group discussion around how to apply TIC concepts to a case [29] Small group discussion around how to apply TIC concepts to a case [29]	Understand the elements of well-being and what allows individuals to flourish Understand character strengths and how they trade off with each other Use character strengths to boost well-being Apply character strengths at work Support consumers to apply their own character strengths Understand and apply TIC concepts in practice Understand mindfulness and associated benefits Apply mindfulness at work Understand explanatory styles Apply optimistic explanatory style at work
5	Mindfulness [42] Optimistic explanatory style [43]	Large group guided meditation [42] Small group discussion about how to apply mindfulness with consumers Seligman’s (2006) Optimism Test Small group discussion around how to apply Seligman’s (2006) ABCDE method with consumers	Understand mindfulness and associated benefits Apply mindfulness at work Understand explanatory styles Apply optimistic explanatory style at work
6	Growth mindset [44] Grit [45]	Self-reflection on mindsets towards the self (Dweck, 2006) Self-reflection on mindsets toward others (Dweck, 2006) Construct a goal hierarchy with most important value as top-level goal [45] Small group discussion around how to connect work with top-level goal	Understand mindsets and their consequences Apply growth mindset at work Apply grit at work Connect and align actions with top-level goal
7	Trauma-Responsive Practice, continued	Large group discussion around how to apply trauma-responsive practices with case Small group discussion around how to apply trauma-responsive practices with case Large group discussion around barriers to applying trauma-responsive practices and solutions for overcoming those barriers	Apply trauma-responsive practice at work effectively
8	Critical consciousness [36,39,46,47,48,49,50,51] Self-Compassion [52] Self-Care [31] Collective Care [31]	Large group brainstorming of critical actionsLarge group discussion around how external forces influence how we think and feel about ourselves and others [31] Large group discussion about how to have strengths-based thoughts about ourselves and others [31] Large group discussion about inhibiting thoughts [31] Large group discussion about how to change inhibiting thoughts [31] Large group discussion about examples of collective care [31]	Understand the definition and consequences of critical consciousness Connect critical consciousness to positive psychological principles and practices Identify targets for critical action Identify critical actions to stop harm Identify critical actions to promote welfare Understand self-compassion and associated benefits Apply self-compassion Understand self-care and associated benefits Apply self-care Understand collective care and associated benefits Apply collective care

## Data Availability

We cannot share access to the data because we did not share any plans to do so with our participants when introducing the evaluation study or with our institutional review board when submitting for exempt status. We also cannot share any pre- and post-survey data that we collected via participant consent and to which we allude in the paper because we did not indicate that we would do so with our participants or with our institutional review board.

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
