# Peer review of "Trauma-Responsive Vocational Rehabilitation Services"

_behavsci, 2023, doi:10.3390/bs13060511_

Round 1

Reviewer 1 Report

Thank you for the opportunity to review this important work. As an aside, I did wonder if you considered mental health as another intersecting limitation in access to VR services. The article feels a little unbalanced with far greater background/lit vs. inquiry methods/results. I would suggest considering reducing the first and building up the second. Thank you for doing this under-studied area and seeking improvements to barriers to access to VR.

Author Response

Thank you for reviewing our manuscript. We appreciate the time, energy, and expertise you donated to help us improve our manuscript. We shifted the balance between the literature review and the methods/results section by trimming down the literature review and using relevant empirical work to discuss themes and recommendations from the focus group data. We hope the changes are to your satisfaction.

Reviewer 2 Report

Congratulations for your contribution to qualitative research and for the context discussed.

The article is in preliminary draft format. I do not know if it is suitable for the journal.

The publication of the empirical phase would be very interesting. In this case, as a preliminary project, publish a pilot study and/or expected results.

Most of the bibliography is more than 5 years old: it is recommended that 50% of the bibliography be from the last 5 years.

Plagiarism reaches 24.73%. It is recommended that it be less than 10%. The introduction and discussion part should be rewritten.

Author Response

Thank you for reviewing our manuscript. We appreciate the time, energy, and expertise you donated to help us improve our manuscript. We addressed your feedback by trimming down the literature review and using relevant empirical work to discuss themes and recommendations from the focus group data. In doing so, we trimmed the references by almost half. We still do not meet the recommendation that 50% of the bibliography be from the last 5 years in large measure because there simply hasn't been a lot of research in this specific area of study, but we are closer now. The remaining references that are older than 5 years are seminal works or the only works available on the particular topics covered that seem essential to the work that we conducted. In trimming the literature review, we also eliminated most of the direct quotations, so the manuscript should no longer exceed 10% plagiarism. We have added language to soften conclusions on pages 18 and 19, recognizing fully the preliminary nature of our findings and the potential benefit of additional programs other than ours to improve VR services. While our paper presents preliminary findings from an initial and uncontrolled pilot, we believe that it adds promising conceptual and empirical content to the literature. We think that there is surely room in the literature, particularly literature that is not more maturely developed, for preliminary and/or pilot projects. We hope the changes are to your satisfaction.

Reviewer 3 Report

I think this programme and the modules presented were excellent. It was so simple but very effective. I think that everyone MUST engage with minorities and see the world (and their traumatic experiences) from their personal point of view. As a South African I can identify with this paper and truly believe that we must think and act from all perspectives, taking personal time to actually to ask everyone if they are "OK" and if they are coping. Well done!  I hope this programme takes off and enables and empowers many of the marginalized people of many, sometimes forgotten communities 

Author Response

Thank you for reviewing our manuscript. We appreciate the time, energy, and expertise you donated to help us improve our manuscript. We are grateful you see value in the work that we're doing to support DVR in improving the quality of their services delivered to low-income, African American consumers. Your positive feedback was well received. 
